# Association between the rs820218 Variant within the *SAP30BP* Gene and Rotator Cuff Rupture in an Amazonian Population

**DOI:** 10.3390/genes14020367

**Published:** 2023-01-31

**Authors:** Rui Sergio Monteiro de Barros, Carla de Castro Sant’ Anna, Diego Di Felipe Ávila Alcantara, Karla Beatriz Cardias Cereja Pantoja, Marianne Rodrigues Fernandes, Lívia Guerreiro de Barros Bentes, Antônio Leonardo Jatahi Cavalcanti Pimentel, Rafael Silva Lemos, Nyara Rodrigues Conde de Almeida, Manuela Rodrigues Neiva Fernandes, Thiago Sequeira da Cruz, Atylla de Andrade Candido, Rommel Mario Rodriguez Burbano

**Affiliations:** 1Hospital Ophir Loyola, Belém 66063-240, Brazil; 2Centro de Ciências Biológicas e da Saúde, Universidade do Estado do Pará, Belém 66087-662, Brazil; 3Rede Mater Dei—Hospital Porto Dias, Belém 66093-020, Brazil; 4Núcleo de Pesquisas em Oncologia, Universidade Federal do Pará, Belém 66073-000, Brazil

**Keywords:** rotator cuff tears, Amazonian population, *SAP30BP*, rs820218, single-nucleotide polymorphism, genetic susceptibility

## Abstract

Background: Rotator cuff disease is one of the leading causes of musculoskeletal pain and disability, and its etiology is most likely multifactorial but remains incompletely understood. Therefore, the objective of this research was to investigate the relationship of the single-nucleotide rs820218 polymorphism of the SAP30-binding protein (SAP30BP) gene with rotator cuff tears in the Amazonian population. Methods: The case group consisted of patients who were operated on due to rotator cuff tears in a hospital in the Amazon region between 2010 and 2021, and the control group was composed of individuals who were selected after negative physical examinations for rotator cuff tears. Genomic DNA was obtained from saliva samples. For the genotyping and allelic discrimination of the selected single nucleotide polymorphism (rs820218) in the *SAP30BP* gene, real-time PCR was performed. Results: The frequency of the A allele in the control group was four times as high as that in the case group (AA homozygotes); an association of the genetic variant rs820218 of the *SAP30BP* gene with rotator cuff tears was not established (*p* = 0.28 and 0.20), as the A allelic frequency is ordinarily low in the general population. Conclusions: The presence of the A allele indicates protection against rotator cuff tears.

## 1. Introduction

Rotator cuff disease is one of the leading causes of musculoskeletal pain and disability. Characteristically progressive, it evolves to ruptures and even degenerative joint processes, causing increasing loss of function that manifests during usual activities as well as work and sports, with significant socio-economic repercussions [1].

The etiology of rotator cuff tearing is most likely multifactorial [2,3,4,5] but remains incompletely understood [1]. Aging is probably the most important patient-related risk factor predisposing individuals to the development of rotator cuff tear (RCT) [6,7,8,9].

Various extrinsic and intrinsic tendon factors have been proposed to explain it. However, there is no consensus regarding the main factors and how they interact [8]. The extrinsic factors include bony impingement and morphologic variations in the coracoacromial arch, such as a curved or hooked acromion and osacromiale. The intrinsic factors include tendon degeneration, tissue hypovascularization, local inflammatory responses, patient comorbidities and genetics [7,8,10].

Familial predispositions to the disease suggest a genetic component in its etiopathogenesis. The history of ruptures among relatives is a risk factor for the occurrence of injuries. Kinship also seems to influence the clinical presentation, including the frequency of symptoms and disease progression [8]. Tashjian et al. have shown strong evidence for a genetic component in RCT, through population genealogy searches linked to medical records [11]. Various genes seem to be involved in the pathophysiology of RCT, especially those involved in extracellular matrix metabolism in the tendons, such as *DEFB1*, *FGFR1*, *FGFR3*, *ESRRB*, *FGF10*, *MMP-1*, *TNC*, *FCRL3*, *SASH1* and *SAP30BP* genes [1,3,8,12,13,14,15,16].

The *SAP30BP* gene is located at 17q25.1 and has 14 exons. There are several genetic variants within this gene, such as rs4453563, rs8076675, rs62090774, rs2898569, rs1661652, rs4999137, rs1661651, rs2053508, rs820218, rs62090776, rs7208873 and s3743999. Studies have identified significant evidence of an association of rs820218 in the SAP30-binding protein (SAP30BP) gene with rotator cuff tears in the American population [17].

There is evidence that the SAP30-binding protein (*SAP30BP* gene) inhibits transcription and induces apoptosis by interacting with the mSin3 complex associated with SAP30-associated mSin3 complex [17,18].

Tashjian et al. documented the results of a genome wide associations (GWAS) on rotator cuff tears conducted to date and identified significant evidence for the association of rs820218 in the *SAP30BP* gene with rotator cuff tears in the American population [16]. More recently, Tian et al. showed that a genetic polymorphism of *SAP30BP* contributes to the risk of rotator cuff tears in individuals with the A allele for SNP rs820218, who were less susceptible to developing rotator cuff tears in a Han Chinese population study [17]. However, the genome-wide association study by Roos et al. did not show a significant association between the *SAP30BP* gene and rotator cuff tears [3].

Even though there is evidence for a positive relationship between variants of the *SAP30BP* gene and RCT, it remains necessary to carry out more studies with diverse ethnic groups, given that each population has its own genetic heterogeneity, in order to identifying patterns regarding variants of the *SAP30BP* gene in these populations affected by rotator cuff disease.

The objective of this research was to investigate the relationship of the single-nucleotide polymorphism (SNP) rs820218 of the *SAP30BP* gene with RCT in the Amazonian population, in order to identify genetic groups at risk, to inform future preventive actions and public health policies.

## 2. Materials and Methods

### 2.1. Subject Selection

All the individuals and patients were included in the study after signing an informed consent document. The project was approved by the Ophir Loyola Hospital Research Ethics Committee under the approval number (4.794.422) and CAAE (44571821.3.0000.5550).

The case group consisted of 100 patients aged between 40 and 80 years with isolated or combined full-thickness tears of the supraspinatus and infraspinatus, operated in a private hospital in the Amazon region between 2010 and 2021. Patients with a history of trauma, diabetes, rheumatoid arthritis, autoimmune diseases, pregnancy, chronic use of systemic steroids, infections, tendinosis only, partial rotator cuff tear or isolated subscapularis tear were excluded.

The control group was composed of 100 individuals between 40 and 80 years of age without shoulder symptoms or functional impairment and selected after a negative physical examination for rotator cuff tears. Patients with a history of trauma, RCT, a family history of shoulder pain or surgery, shoulder surgery, diabetes, rheumatoid arthritis, autoimmune diseases, pregnancy, chronic use of systemic steroids, infections, tendinosis only, a partial rotator cuff tear or an isolated subscapularis tear were excluded. The baseline demographic and clinical characteristics of both groups are described in Table 1. In total, 79 individuals in the control group and 78 patients in the case group were considered as valid samples for genetic analysis.

Therefore, the sample of the present study is representative of the Amazonian population, although not large, and comprised eligible patients at the reference hospital over a decade. The number of patients in the control group was selected with a view to matching based on clinical data.

Clinical–epidemiological data were collected from both groups regarding age, gender, self-reported ethnicity, smoking, alcohol consumption, autoimmune diseases, hypothyroidism, hypertension, tendinopathies in areas other than the shoulder and work or sports practices involving upper limb elevation.

### 2.2. DNA Collection

The DNA of the participants in both the case and control groups was collected from buccal mucosa samples. The cells were collected by scraping the oral mucosa using a buccal swab (swab) using the 4N6FLOQSwabsTM model (regular tip, peel pouch), according to Kücheleret al.’s method [19].

The samples were stored and processed in the Molecular Biology Laboratory of our institution. The DNA extraction was performed using the PureLinkTM Genomic DNA Mini Kit, according to the manufacturer’s protocol (Qiagen, Hilden, Germany).

### 2.3. Single-Nucleotide Polymorphism Selection and Genotyping

For the genotyping and allelic discrimination of the selected SNP (rs820218) in the *SAP30BP* gene, real-time PCR was used with specific probes for the TaqMan^®^ SNP Genotyping Assay (Applied Biosystems, Foster City, CA, USA) in the QuantStudio 6 Flex Real- time PCR system. The TaqManGenotyper software was used for data analysis, determining the genotype read accuracy and genotyping quality control.

### 2.4. Statistical Analysis

Hardy–Weinberg equilibrium (HWE) tests were conducted within the control group for the SNP (rs820218) to determine whether the populations were in equilibrium. The clinical and epidemiological data were analyzed using the Fisher exact and 𝜒2 tests. Frequency analyses of genotypes and alleles were performed using a multivariate logistic regression, and the test was controlled for clinically and epidemiologically significant variables in order to avoid misinterpretation of the results. The risks associated with individual alleles and genotypes were calculated as the odds ratios (ORs) with 95% confidence intervals (CIs).

Furthermore, a comparison between the allelic frequencies of the SNP rs820218 of the *SAP30B* gene of the case group was performed with the five continental populations available in the 1000 Genomes Project (http://www.1000genomes.org/ accessed on 1 August 2022), with the objective of verifying whether there would be a difference between the populations. The data available in the 1000 Genomes database are public and formed from the genetic information of 5 different populations: 661 individuals from Africa (AFR), 503 from Europe (EUR), 347 from Americas (AMR), 504 from East Asia (EAS) and 489 from South Asia (SAS) (Table 2). Thus, Fisher’s exact test was employed to compare the allelic frequencies of the investigated variant in two-by-two groups and obtain the *p* value. Values of *p* < 0.05 were considered statistically significant. All the analyses were performed using the software R v.4.0.5 (The R Foundation, 2016).

## 3. Results

The clinical and epidemiological variables were compared between the case and control groups and are shown in Table 1. Age (*p* value = 0.0011), hypertension (*p* value = 0.001) and tendinopathies (*p* value = 0.002) were statistically significantly different between groups. The distributions of the variables that were statistically significant, added to the variable “Overhead activities”, are represented in Figure 1.

The association of the genetic variant rs820218 of the *SAP30BP* gene with the RCT was investigated, and a statistically significant result was not obtained (*p* = 0.28 and *p* = 0.20). However, we can observe that the frequency of the A allele in the control group is four times as high as that in the case group (AA homozygotes), suggesting that the A allele provides protection against RCT (Table 2).

We also compared the allelic frequencies of the case group sample (RCT), obtained from the gene count, with the allelic frequencies of the 1000 Genomes populations. However, we did not find any statistically significant results (Table 3 and Table 4).

## 4. Discussion

This study contributes to a better understanding of rotator cuff tears, which are among the most frequent injuries of the musculoskeletal system, along with injuries in the lumbar and cervical regions. RCT can lead to a loss of shoulder function to varying degrees. The complications of RCT include tendinitis and partial and total tears of the cuff tendons. It is noteworthy that the sample used in this research reflects the reality of the Amazonian population, with this study being the first study carried out in the region. In this context, we emphasize the importance of developing multicenter studies in order to expand the sample size and reinforce the findings of this research.

Our results showed statistically significant differences between the case and control groups in the following variables analyzed: age, hypertension and tendinopathies. Aging is considered a major risk factor for RCTs, and our results indicate that it is related to a higher probability of their occurrence [20,21]. Hypertension was another significant epidemiological factor identified in the case group compared to the control, which was also observed in other studies [20,22]. Other studies did not find a relationship with systemic arterial hypertension [23,24].

Another study concluded that genetic polymorphisms in MMP-1 and MMP-3 were associated with rotator cuff tears. Individuals with the haplotype 2G/5A were more susceptible to RCTs in the population studied, where collagen degeneration and the disordered arrangement of collagen fibers in RCT are associated with an increase in the activity of matrix metalloproteases 1 and 3 (MMP-1 and MMP-3), and that MMP activity may be, in part, genetically modulated. The groups did not differ regarding race, smoking and the presence of high blood pressure (*p* = 0.692, *p* > 0.999 and *p* = 0.831, respectively). Tendinopathy in other locations was more prevalent among patients with RCT (*p* = 0.016). However, risk factors such as gender [25], smoking [26], professional activity with repetitive shoulder efforts [27] and sports using the upper limbs [28] did not differ between the two groups studied, thus corroborating our results.

No significant differences were observed regarding variables such as gender and work or sports activity using the shoulders excessively; similar results were observed in the study by Tian et al. [17].

We found that the frequency of the A allele in the control group was four times as high as that in the case group (AA homozygotes), while an association of the genetic variant rs820218 of the *SAP30BP* gene with rotator cuff tears was not established (*p* = 0, 28 and 0.20). This is because the A allelic frequency is ordinarily low in the general population, according to several studies [3,7,10,16,17]. It is worth mentioning the study by Roos et al. [3], who performed a genome-wide association screening with public domain data (Research Program in Genes, Environment and Health), involving 8357 patients with RCTs and 94,622 control individuals, and demonstrated that the frequency of the A allele in the study population was low. Even with the large sample size used, they did not observe statistical significance for the association of the genetic variant rs820218 of the *SAP30BP* gene with RCT, corroborating the findings of the present study.

It is important to emphasize the great difficulty in replicating and applying the findings, mainly due to the lack of statistical power, the high rate of false-positive results and the large number of variables involved. In addition, diseases are influenced by the sum of the expression of several genes, but each one of them has a small effect. It is still necessary to catalog different polymorphisms related to RCT, since genomic profiling will allow the establishment of a bank of genetic markers that may contribute to predicting the risk of the disease [29].

As one can observe, the frequency of the variant in the case group is similar to the frequencies of the EAS (East Asia) and SAS (South Asia) populations obtained from the 1000 Genomes database (Table 3). This similarity supports the hypothesis that Asian people migrated to America through the Bering Strait. This hypothesis proposes that the composition of the first American inhabitants would have been established by the migration of Asian populations through a land bridge linking Northeast Asia and North America [30]. As the investigated population is part of a highly mixed population and due to the large contribution of Amerindian populations to its composition [31], such similarity was to be expected. Although no statistically significant difference was found between the investigated population and the five continental populations of the 1000 Genomes Project, we can observe that the European population is the most divergent. Due to the fact that the A allele is more frequent in the European population, a lower prevalence of RCT could be expected, but more studies are necessary to confirm this hypothesis. Therefore, we suggest that the A allele provides protection against RCT, but it is important to be aware that RCT is a multifactorial disease influenced by genetics.

This finding supports a recent study performed in a Chinese population, in which 394 patients with RCT and 998 healthy controls were included, showing that individuals with the A allele for the SNP rs820218 were less susceptible to the development of rotator cuff tears [17].

Tashjian et al. demonstrated a genome-wide association for RCT, with two significantly associated genetic variants: *SAP30BP* (rs820218) on chromosome 17q25 and *SASH1* (rs12527089) on chromosome 6q24. Both genes have been reported to be involved in overall cell death or apoptosis; abnormalities within these genes can increase their activity, leading to increased apoptosis of tendon cells, predisposing individuals to RCT [16].

In this context, the SAP30BP-binding protein is a transcriptional regulatory protein located on chromosome 17 that is constantly present in musculoskeletal tissue. Its primary function is to induce cell death or apoptosis, and it may act as a transcriptional corepressor of a gene related to cell survival [18]. SAP30BP could also potentially be involved in increasing tendon apoptosis in the rotator cuff tear scenario [16].

As the *SAP30BP* gene is related to transcription inhibition and apoptosis induction, SNPs may provide new information about differences between individuals in the development of ruptures and reveal insights into the underlying mechanisms of shoulder tendinopathy [32].

Sejersen et al. [33], in a systematic review of the literature, identified 2199 studies involving protein analysis in tendinopathies and found a trend of increased expression of COL1, COL3, MMP1, MMP9, MMP13, TIMP1 and VEGF and decreased MMP3 [33], demonstrating the importance of cataloging different polymorphisms related to RCT, which may contribute to a better understanding of the lesion.

In another line of investigation seeking to elucidate whether fatty infiltration and the inflammatory process delayed healing in RCT, Thankam et al. evaluated miRNAs from RCT patients with and without IG and inflammation and detected 13 miRNAs and 216 target genes that interconnected the metabolic checkpoint 5′ monophosphate-adenosine activated protein kinase (AMP-Q) and the TREM-1 inflammatory molecule pathway [34].

## 5. Conclusions

The rs820218 variant of the *SAP30BP* gene does not contribute to the risk of rotator cuff tears in the present sample of the Amazonian population. Despite this, the findings suggest that individuals with the A allele have a tendency to be protected against rotator cuff tears. We also observed statistically significant differences between the case and control groups in the following analyzed variables: age, hypertension and tendinopathies.

## Figures and Tables

**Figure 1 genes-14-00367-f001:**
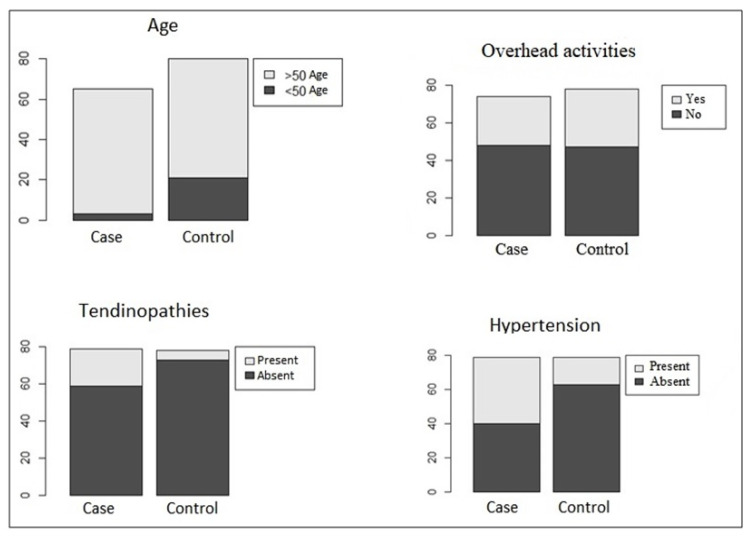
Distribution of clinical-epidemiological variables that where statistically significant with the variable “overhead activities”.

**Table 1 genes-14-00367-t001:** Clinical–epidemiological variables of the case and control groups.

Variable	Case	Control	*p* Value ^1^
Age			**0.0011**
>50 years	3 (4.6%)	21 (26.2%)	
<50 years	62 (95.4%)	59 (73.8%)	
Gender			0.6283
Female	46 (57.5%)	50 (62.5%)	
Male	34 (42.5%)	30 (37.5%)	
Self-reported ethnicity			0.2639
Yellow	2 (2.5%)	3 (3.9%)	
White	29 (36.3%)	27 (35.1%)	
Brown	46 (57.5%)	38 (49.3%)	
Black	3 (3.7%)	9 (11.7%)	
Overhead activities			0.675
No	48 (64.8%)	47 (60.3%)	
Yes	26 (35.2%)	31 (39.7%)	
Smoking			0.532
No	76 (95%)	73 (91.2%)	
Yes	4 (5%)	7 (8.8%)	
Alcoholism			0.532
No	76 (95%)	73 (91.2%)	
Yes	4 (5%)	7 (8.8%)	
Hypothyroidism			1.000 ^2^
Absent	74 (96.1%)	76 (95%)	
Present	3 (3.9%)	4 (5%)	
Hypertension			**0.001**
Absent	40 (50.6%)	63 (79.7%)	
Present	39 (49.4%)	16 (20.3%)	
Tendinopathies			**0.002**
Absent	59 (74.7%)	73 (93.6%)	
Present	20 (25.3%)	5 (6.4%)	

^1^ 𝜒2 tests; ^2^ Fisher exact test.

**Table 2 genes-14-00367-t002:** Association of the rs820218 variant of the *SAP30BP* gene with the case and control groups.

Genotype	Case N (%)	Control N (%)	OR	CI95%	*p* Value
GG ^1^	44 (73.3%)	43 (55.1%)	1.00		0.2821
AG	12 (20.0%)	19 (24.4%)	0.64	0.26–1.61	
AA	4 (6.7%)	16 (20.5%)	0.40	0.11–1.44	
GG ^2^	44 (73.3%)	43 (55.1%)	1.00		0.1464
AG + AA	16 (26.7%)	35 (44.9%)	0.55	0.25–1.24	
GG + AG ^3^	56 (93.3%)	62 (79.5%)	1.00		
AA	4 (6.7%)	16 (20.5%)	0.46	0.13–1.59	0.2023

The significant variables—age, hypertension and tendinopathies—were controlled for in the analyses. These variables were included to adjust the OR in the allelic model of rs820218. ^1^ Codominant genotype; ^2^ Dominant genotype; ^3^ Recessive genotype.

**Table 3 genes-14-00367-t003:** Distributions of allelic frequencies of rs820218 of the *SAP30BP* gene in the populations.

Allele (rs820218)	CASE	AFR	AMR	EAS	EUR	SAS
G	0.83	0.79	0.79	0.84	0.65	0.81
A	0.17	0.21	0.21	0.16	0.35	0.19

Africa (AFR), Europe (EUR), America (AMR), East Asia (EAS) and South Asia (SAS).

**Table 4 genes-14-00367-t004:** Comparison of allelic frequencies between the case population and each of the 1000 Genomes populations.

	Pairwise Comparison (*p* Value)				
Variant	CASE × AFR	CASE × AMR	CASE × EAS	CASE × EUR	CASE × SAS
rs820218	1.00	1.00	1.00	0.21	1.00

Africa (AFR), Europe (EUR), America (AMR), East Asia (EAS) and South Asia (SAS).

## Data Availability

The data presented in this study are available on request from the corresponding author. The data are not publicly available due to the privacy terms contained in the informed consent agreement.

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
