# Peer review of "Association between the rs820218 Variant within the SAP30BP Gene and Rotator Cuff Rupture in an Amazonian Population"

_genes, 2023, doi:10.3390/genes14020367_

Round 1

Reviewer 1 Report

In this study, the authors investigated SAP30BP gene polymorphism (rs820218) and the risk of rupture of the rotator cuff in the Amazon population. Although this topic is interesting, and the study is well-designed, I have several concerns that should be addressed, as follows:

  • Introduction:
    • Authors should mention the location of the gene encoding SAP30BP.
    • There are several genetic variants of SAP30BP. Please describe them.
  • Methods:
    • The authors investigated only 79 cases and 78 controls, which appear to be relatively small samples. Please add a justification for the sample size used.
    • DNA was extracted from saliva samples. Please explain why you used saliva samples and not blood samples. And the method used for sampling—scraping the oral mucosa using a buccal 104 swab—denotes that the sample was not saliva.
  • Results:
    • Many significant factors in Table 1 could affect the occurrence of RCT, such as age, hypertension, and tendinopathies, which are estimated in cases and controls. However, they should be used to adjust the OR of the SNP.
    • A multiplicative genetic model (allelic model) should be assessed in table II.
    • Figure 1 is redundant as it shows the same results as Table I.
    • In table I, please revise the writing style for gender.
    • Hardy-Weinberg equilibrium should be tested in the control group.
  • Throughout the manuscript, there are several structural and grammatical errors. The whole manuscript should be carefully revised by a native English speaker.

Reviewer 2 Report

In this study the authors analyze the eventual association of SNP rs820218 in the SAP30BP gene with rotator cuff tears (RCT) in the Amazonian population. The aim of the study is interesting, and the real-time PCR is a suitable method. However, I have some major concerns regarding the experimental design.

First, 78 patients affected with RCT and 79 healthy controls is a relatively small sample for a case-control association study.

Second, the patient group and the control group were not age matched (with maximum difference in age not more than 2 years). Thus, since rotator cuff tears show an age-related progression some biases in the results may arise from the inclusion of older patients who have been exposed to environmental factors for more years than younger control cases.
Moreover, results regarding the percentage of patients with age >50 (4.6%) versus 95.4% of patients with age < 50 as reported in Table 1 are contrasting with the percentage shown in figure 1, which appear to be reversed.

The same mistake occurred for the control group.

Third, no functional studies have been carried out to verify the variation of expression of the SAP30BP gene at mRNA or protein level in individuals with the A allele for SNP rs820218.

Some parts of the Discussion section (lines 222-228 and lines 231-233) are too speculative.

Round 2

Reviewer 1 Report

The authors have adequately addressed all my concerns

Reviewer 2 Report

No comments